# Numerical Study of the Internal Fluid Dynamics of Draft Tube in Seawater Pumped Storage Hydropower Plant

**Jianyong Hu [1,2], Qingbo Wang [1,3], Zhenzhu Meng [2,4,\*], Hongge Song [1,3], Bowen Chen [5] and Hui Shen [1,6]**

[1]   School of Geomatics and Municipal Engineering, Zhejiang University of Water Resources and Electric Power, Hangzhou 310018, China; hujy@zjweu.edu.cn (J.H.); wqb5113@163.com (Q.W.); shg1069876245@163.com (H.S.); huishen0807@163.com (H.S.)
[2]   Engineering Research Center of Digital Twin Basin of Zhejiang Province, Hangzhou 310018, China
[3]   College of Energy and Power Engineering, North China University of Water Resources and Hydropower, Zhengzhou 450045, China
[4]   School of Water Conservancy & Environment Engineering, Zhejiang University of Water Resources and Electric Power, Hangzhou 310018, China
[5]   School of Civil and Hydraulic Engineering, Huazhong University of Science and Technology, Wuhan 430000, China; 13083603612@163.com
[6]   School of Water Resources and Hydropower, Hebei University of Engineering, Handan 056038, China
\*   Correspondence: mengzhzh@zjweu.edu.cn

**Abstract:** Pumped storage hydropower plants are renewable energy systems that are effective in saving energy and solving electricity peak-on shortage. Seawater pumped storage hydropower plants are a novel type of pumped storage hydropower plant specifically supplying electric power for ocean islands with the support of solar energy and wind energy. Compared with traditional pumped storage hydropower plants that are constructed on the mainland, seawater pumped storage hydropower plants should take the influence of the complex marine environment, such as extreme waves and winds, into consideration. Taking the characteristics of waves near islands in the East China Sea as an example, we explored the transient hydraulic characteristics in the draft tube of a pump turbine under wave disturbance using a sliding grid interface and the detached eddy simulation (DES) turbulence model. By analyzing the characteristics of unsteady flow in the draft tube, the vortex characteristics under the Q criterion, the frequency characteristics of the pressure pulsation, the evolution law of the internal fluid, and the propagation law of the pressure pulsation were explored. For the situation without wave disturbance, an obvious eccentric vortex in the straight cone section of the draft tube was observed in the case where the opening of the guide vane was small. With the increase in the opening of the guide vane, the eccentric vortex gradually dissipated. For the situation with wave disturbance, the main frequency of the draft tube equaled the frequency of the wave disturbance, the maximum pressure pulsation at the selected monitoring points increased 5 to 15 times, and the superposition of the wave pressure pulsation signals and the draft tube pressure pulsation produced more low-frequency, high-amplitude pressure pulsation signals. Even though the pressure pulsation frequency spectrum varied a lot, the frequency domain of the pressure pulsation without wave disturbance still existed. In addition, the wave disturbance merely varied with the pressure of the draft tube. The influence of wave disturbance on the pressure distribution in the draft tube was relatively small. The results can provide a reference for the operation of seawater pumped storage hydropower plants.

**Keywords:** wave disturbance; pump turbine; guide vane opening; draft tube; pressure pulsation

## 1. Introduction

Pumped storage hydropower plants are renewable energy systems that aim to solve electricity peak-on shortage problems. Seawater pumped storage hydropower plants are a novel type of pumped storage hydropower plant that are constructed with the support

of solar energy and wind energy. Seawater pumped storage hydropower plants are used to supply electric power for ocean islands, which are difficult to connect with mainland electricity networks. Under the background of China's efforts to achieve the goal of "peak carbon dioxide emissions and carbon neutrality", the development of the pumped storage industry has ushered in a new stage. Seawater pumped storage hydropower plants are a novel type of pumped storage hydropower plant, which can not only form a stable power supply system with wind energy and solar energy but also solve the fresh water supply problem of ocean islands. According to the latest census reports of seawater pumped storage hydropower plant data released by the Chinese National Energy Administration, 238 points that offer the possibility of constructing seawater pumped storage hydropower plants have been discovered. The potential total capacity could reach 42.083 million kilowatts [1,2]. Similar to conventional pumped storage units, seawater pumped storage units undertake the task of load and frequency control. As the hydraulic unit output power varies frequently, engineers often change the guide vane opening to adjust the volume of flow to adjust the output power [3,4]. However, the difference between seawater pumped storage hydropower plants and conventional pumped storage hydropower plants is that the lower reservoir of seawater pumped storage is located in the marine environment. Many complicated influencing factors including typhoons, tides, and ocean currents may induce rapid changes in the water level, which may increase the operating range of the unit head and accordingly increase the possibility of hydraulic instability in the pump turbine [5–7].

When the pump turbine deviates from the optimal working condition, the draft tube pressure pulsation is one of the main factors affecting the stable operation of the pump turbine. The pressure pulsation of the draft tube is mainly induced by two factors: one is the dynamic and static interference of the runner; the other is the secondary flow in the draft tube. The pressure pulsation caused by the dynamic and static interference mainly includes the frequency conversion, blade frequency, and double frequency of the blade frequency. The pressure pulsation resulting from the secondary flow has the characteristics of a low frequency and high amplitude [8–12]. Previous studies mainly provided insights into the internal dynamics of draft tubes considering different types of units and levels of opening of the guide vanes. Taking a typical partial load condition of a Francis turbine as an example, Zhou et al. simulated the velocity field of the vortex rope in a draft tube. They also predicted the frequency of pressure pulsation in the vortex rope and determined the pressure pulsation on several different sections of the draft tube [13]. Ji et al. simulated the unsteady flow in the draft tube of a Francis turbine and found that a vortex rope with a low pressure and large flow rate may lead to a low-pressure pulsation in the turbine [14]. Gao et al. explored the internal flow dynamics of a tidal turbine with an indicated guide vane opening of the tubular unit and proposed that the rotating flow in the draft tube under low pressure is mainly caused by the air mass and hydraulic imbalance at the outlet of the runner; thus, the rotating flow cannot be simply defined as the vortex rope [15]. Ji et al. pointed out that the large and invisible vortex in the conical cross-section at the inlet of the draft tube gradually changes to a tangible vortex rope with an increase in the guide vane opening [16]. Wang and Cavazzini simulated the pressure fluctuation in the S-shaped region of a pump turbine and captured the low-frequency component caused by the rotating stall phenomenon at a low flow rate, which was induced by the rotating stall phenomenon [17,18]. By analyzing the pressure difference between the inner wall of the draft tube center and the inner wall, Yang et al. determined the relation between the pressure pulsation values of the inner wall and that of the center wall of a draft tube [19]. Guo et al. found that the smaller the opening of the guide vane, the more disordered the vortex generated by each part of the turbine, and the more unstable the flow field. The original shape of the vortex rope in the draft tube is disturbed due to the interaction between vortices, which affects the efficient and stable operation of the turbine [20]. Chen et al. analyzed the influence of upstream disturbance on the draft tube flow of a Francis turbine under partial load conditions using different vortex identification methods [21].

The results indicated that when the guide vane opening is small, the upstream disturbance propagates toward the downstream, which has a significant impact on the flow pattern of the downstream draft tube. When the opening approaches its optimal status, the upstream disturbance becomes weak, and thus the associated influence on the downstream flow pattern is fairly limited.

Numerous numerical simulations of the fluid dynamics of steady and unsteady flows of conventional pumped storage units have been conducted [22,23]. However, studies on wave disturbance in the upstream and downstream of these units have mostly focused on low-head tubular turbines whose water head and output are relatively small [24,25]. Different from the low-head and small-capacity tubular turbines that are commonly installed in conventional pumped storage hydropower plants, seawater pumped storage hydropower plants commonly adopt a vertical-shaft Francis turbine with a high head and large capacity, which has a high possibility of becoming hydraulically unstable when facing the disturbance of periodic waves. Therefore, this study aimed to reveal the influence of the disturbance of periodic waves on a pump turbine using numerical simulation, by exploring the evolution law of the flow field and the propagation law of the pressure pulsation in the draft tube. The laws and mechanisms obtained from the results can provide a reference for maintaining the operating stability of seawater pumped storage units.

## 2. Simulation Configuration

### 2.1. Model Parameters and Computing Domain

Using Fluent, we conducted a numerical simulation of a three-dimensional full channel of a pump turbine, which mainly includes five areas: spiral case, stay vanes, guide vanes, runner, and draft tube (see Figure 1). In order to accurately capture the characteristics and propagation law of the pressure pulsation in the draft tube, 9 monitoring points were arranged at every 0.1 m from the center of the draft tube to the bend section, which were labeled P1, P2, P3, P4, P5, P6, P7, P8, and P9. The initial settings of the parameters of the modeled pump turbine under the condition of a hydraulic turbine are shown in Table 1. The calculation was carried out using a rack server with 64-bit high-performance processors, a 36-core single-threaded Intel Xeon Gold 6140 CPU, and a memory of 512 GB.

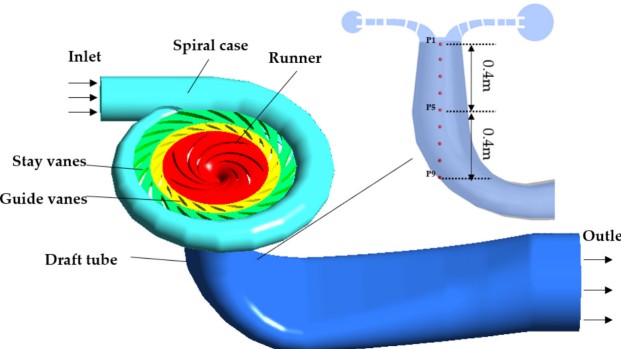

**Figure 1.** Three-dimensional fluid domain model of the pump turbine.

**Table 1.** Parameters of the defined pump turbine.

| Parameters | Initial Settings |
|---|---|
| Runner inlet diameter $D_1$ | 552.2 mm |
| Runner outlet diameter $D_2$ | 250 mm |
| Number of runner blades | 7 |
| Number of stay vanes | 20 |
| Number of guide vanes | 20 |
| Rated head | 32 m |
| Rated speed | 1000/min |

### 2.2. Grid Division

In this study, an unstructured grid was used to divide the flow components. The grid division was conducted using Fluent Meshing. The detached eddy simulation (DES) turbulence model was adopted, in which the grid was divided into a viscous bottom layer, and the boundary of the viscous bottom layer was considered to be y+ less than 10. To improve the accuracy of the calculation, the boundary layer of the straight cone section of the runner and draft tube was locally encrypted to ensure that its average y+ was less than 10. As shown in Figure 2, we selected the steady-state calculation of three-dimensional turbulence under the rated conditions and verified the grid independence based on the parameters of efficiency. In the case of the number of grids larger than 4.5 million, the improvement in efficiency was neglectable with the increase in the grid number. We thus defined 5 million grids in the present case study.

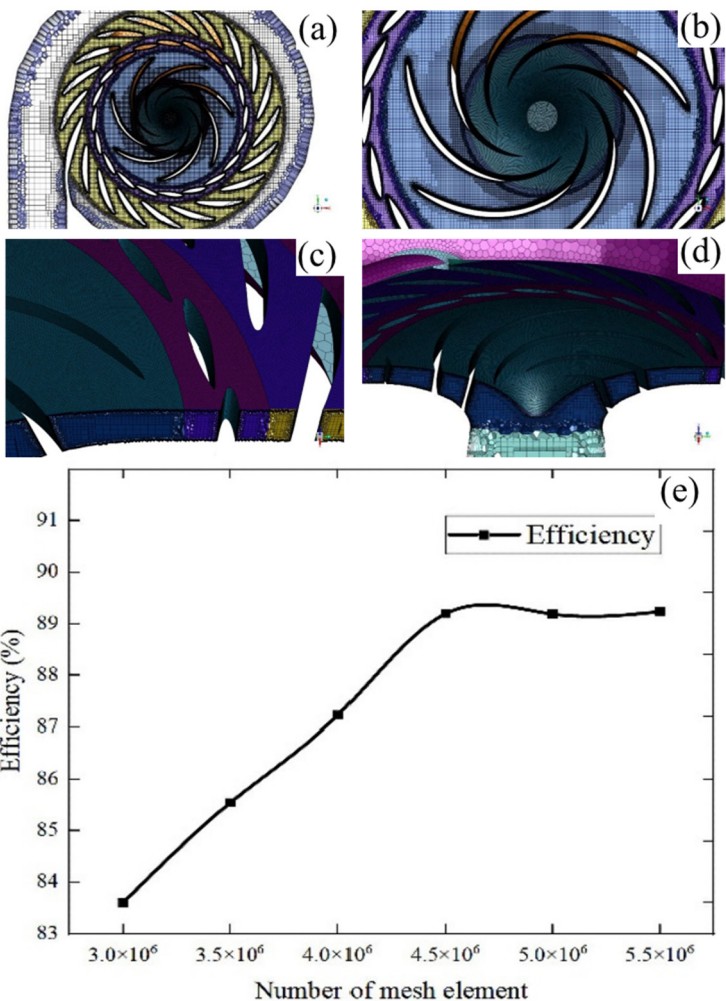

**Figure 2.** Grid division and grid independence verification: (**a**) the local grid division of spiral case cross section, (**b**) the local grid division of runner cross section, (**c**) the local grid division of stay vanes and guide vanes, (**d**) the local grid division of runner longitudinal section, (**e**) verification of grid number independence.

### 2.3. Turbulence Model and Boundary Conditions

As a type of rotating fluid machinery, pump turbines demonstrate the phenomenon of flow separation in the runner area. Previous studies have demonstrated the high accuracy and stability of the detached eddy simulation (DES) turbulence model in solving large separation problems of fluid machinery [18]. The DES model has the advantages of both the large eddy simulation (LES) model and the Reynolds-averaged Navier–Stokes (RANS)

model; thus, it can solve both large eddy structures far away from the wall using the LES model and the turbulent flow near the wall using the RANS model. Thus, the DES turbulence model was used in the present numerical simulation. The numerical model of unsteady flow was initially established and modified on the basis of the governing equations of steady flow. The time-averaged continuity equation and the Reynolds equation of turbulent flow are as follows:

$$\frac{\partial \rho}{\partial t} + \frac{\partial}{\partial x_i}(\rho u_i) = 0 \tag{1}$$

$$\frac{\partial}{\partial t}(\rho u_i) + \frac{\partial}{\partial x_j}(\rho u_i u_j) = -\frac{\partial p}{\partial x_i} + \frac{\partial}{\partial x_j}\{\mu\frac{\partial u_i}{\partial x_j} - \rho\overline{u'_i u'_j}\} + S_i \tag{2}$$

$$\frac{\partial}{\partial t}(\rho\varphi) + \frac{\partial}{\partial x_j}(\rho u_j \varphi) = \frac{\partial}{\partial x_j}\{\Gamma\frac{\partial \varphi}{\partial x_j} - \rho\overline{u'_j \varphi}\} + S \tag{3}$$

$$\frac{\partial}{\partial t}(\rho k) + \frac{\partial}{\partial x_j}(\rho k u_i) = \frac{\partial}{\partial x_j}[(\mu + \frac{\mu_t}{\sigma_k})\frac{\partial k}{\partial x_j}] + u_t(\frac{\partial u_i}{\partial x_j} + \frac{\partial u_j}{\partial x_i})\frac{\partial u_i}{\partial x_j} - \rho C_D\frac{k^{3/2}}{l} \tag{4}$$

where $\rho$ is the fluid density, $u$ is the fluid velocity, $\mu$ is the fluid dynamic viscosity, $\Gamma$ is the fluid parameters, and $S$ is the source term.

For the boundary conditions of the numerical model, the spiral case inlet was set as the mass flow inlet, and the draft tube outlet was set as the pressure outlet. The SIMPLEC algorithm was used to couple the pressure and velocity. The standard discrete format was used to describe the pressure term. The second-order upwind format was used to describe the momentum and turbulent kinetic energy dissipation rate. The solid wall was set to the condition of a no-slip wall, and the convergence accuracy was set to $10^{-5}$. The calculation for a steady flow was firstly carried out. Then, according to the obtained stable macro parameters, the result of the steady-flow calculation was taken as the initial condition of the unsteady-flow calculation. It was verified in the calculation that the convergence accuracy could be met in 30 steps once the iterative convergence of the unsteady-flow calculation was stable. In order to ensure the numerical accuracy and save calculation time, the maximum number of iterations in each time step was set to 30, and the time step was set to 0.0005 s (rotate 3° for each time step). The total calculation time was 10 s with 166 cycles.

*2.4. Selection of Working Conditions at Calculation Points*

In this paper, the flow field in the first quadrant of the pump turbine was studied. The selected working points are shown in Table 2. The water head in the table was 32 m. The wave disturbance at the outlet of the draft tube used the wave data of an island in the East China Sea from the last ten years. In line with Yu et al. [26], the waveform was a sine wave, the maximum effective wave height was 10 m, and the period was 6 s. According to micro-amplitude wave theory [27], the expression of pressure at any point in the wave field can be expressed as

$$p = \rho g(\eta K_Z - Z) \tag{5}$$

where $p$ is the pressure, $\rho$ is the fluid density, $\eta$ is the instantaneous vertical distance of the free surface above the average water surface, $K_z$ is the pressure sensitivity coefficient, which decreases with the increase in the distance between the particle position and the still water surface, and $z$ is the elevation of the free surface.

**Table 2.** Working conditions at the calculation points.

| Working Condition | Relative Opening of Guide Vane is $\alpha_0$ (Degrees) | Unit Speed $n_{11}$ (r/min) | Unit Flow $Q_{11}$ (m$^3$/s) | Outlet Pressure, Pout (Pa), of Draft Tube |
|---|---|---|---|---|
| Hydraulic turbine operating condition (Without wave disturbance) | 6 | 40 | 0.19542 | 0 |
| | 12 | 40 | 0.41603 | 0 |
| | 24 | 40 | 0.65730 | 0 |
| Hydraulic turbine operating condition (wave disturbance) | 6 | 40 | 0.19542 | $10{,}000 \ \sin \frac{\sqrt{10}\pi t}{3}$ |
| | 12 | 40 | 0.41603 | $10{,}000 \ \sin \frac{\sqrt{10}\pi t}{3}$ |
| | 24 | 40 | 0.65730 | $10{,}000 \ \sin \frac{\sqrt{10}\pi t}{3}$ |

Therefore, the variation in hydrostatic pressure and hydrodynamic pressure can be regarded as a sine or cosine function. For small-amplitude waves, we assume that the hydrostatic pressure and hydrodynamic pressure are related to the water depth. According to [28], the formulas for the wave scale are as follows:

$$\lambda_{\mathrm{p}} = \lambda_k = \frac{l_p}{l_m} \tag{6}$$

$$\lambda_{\mathrm{f}} = \lambda_k{}^{-\frac{1}{2}} \tag{7}$$

where $\lambda_k$ is the model length scale, $l_p$ is the prototype length, $l_m$ is the model length, $\lambda_p$ is the pressure scale, and $\lambda_f$ is the frequency scale.

In this paper, the ratio between the prototype and real pump turbine was one to ten. The downstream water level fluctuation was $\Delta H = 1 \times \sin \frac{2\pi t}{T}$. The draft tube outlet pressure fluctuation $\Delta P = 10{,}000 \ \sin \frac{\sqrt{10}\pi t}{3}$ was taken as the wave disturbance condition. Without wave disturbance, the data points in 10 rotating periods (0.6 s) of the runner were analyzed. With wave disturbance, the data points in a sinusoidal period (about 1.9 s) of the wave were analyzed.

## 3. Validation of Simulation Method

In order to obtain the evolution of the pressure pulsation in the draft tube of the model pump turbine, three operating points of the guide vane opening were simulated. Figure 3 compares the experimental results and numerical simulation results for the case of no wave disturbance. The unit rotating speed and unit discharge are given by Equations (8) and (9), respectively [29]:

$$n_{11} = \frac{nD_2}{\sqrt{H}} \tag{8}$$

$$Q_{11} = \frac{Q}{D_2{}^2 \sqrt{H}} \tag{9}$$

where $n_{11}$ is the unit rotating speed, $Q_{11}$ is the unit discharge, $D_2$ is the diameter of the runner outlet, and $H$ is the hydraulic turbine head.

It can be seen that the simulation results fit fairly well with the experimental results. The errors of both the unit speed $n_{11}$ and unit flow $Q_{11}$ are within 3%. This indicates that the numerical simulation method used in this study is effective and reliable.

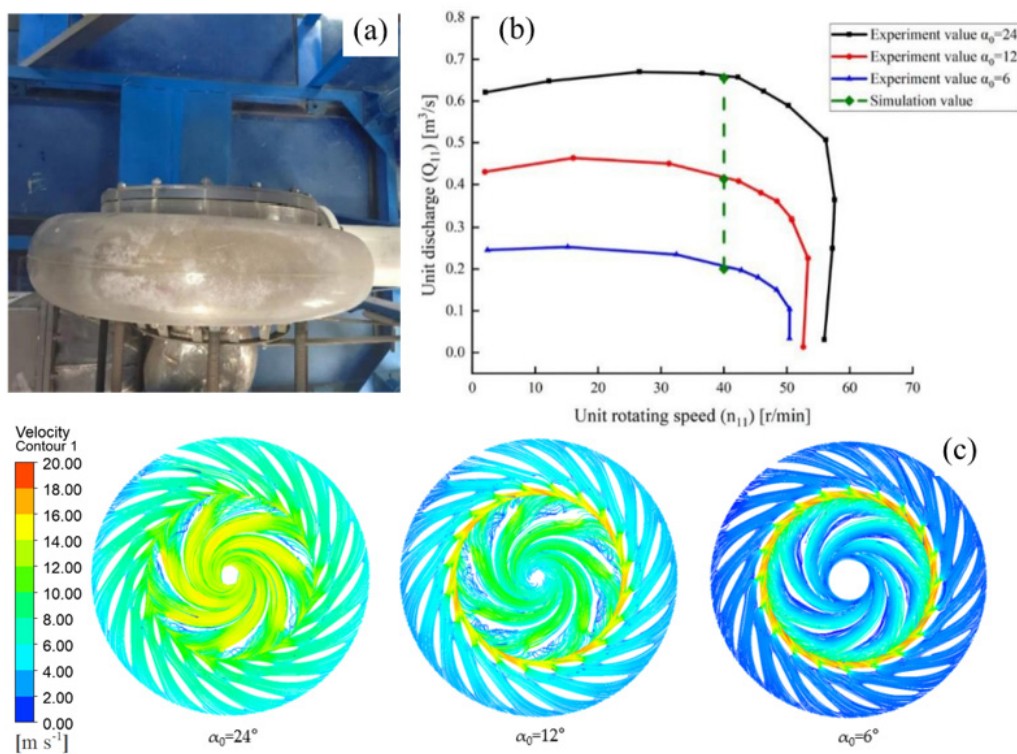

**Figure 3.** Comparison of simulated and experimental data: (**a**) the testbed of the Francis pump turbine; (**b**) the variation in the unit discharge against the unit rotating speed; (**c**) the velocity field of the pump turbine with different guide vane openings.

## 4. Results and Discussion

### 4.1. Pressure and Velocity Distribution in Draft Tube

As shown in Figure 4, for several operating conditions without wave disturbance, a large range of negative pressure is observed in the central area of the straight cone section of the draft tube. At $\alpha_0 = 6°$ and $\alpha_0 = 12°$, the fluid speed of the draft tube inlet is unbalanced. In addition, an obvious pressure gradient is observed in the central area of the straight cone section of the draft tube, and the highest flow rate is 13.4 m/s at $\alpha_0 = 6°$. With the increase in the flow rate, the circumferential velocity component produced by the outlet speed of the runner decreases at $\alpha_0 = 24°$, with the maximum flow rate reducing to 6.66 m/s. With the increase in the loss of steady water, the pressure gradient at the inlet of the draft tube disappears, the negative pressure area in the draft tube decreases, and the maximum pressure appears at the elbow of the draft tube.

Figure 5 shows the static pressure distribution on the axial surface of the draft tube in a wave cycle under the running condition of $\alpha_0 = 6°$. Due to the influence of wave disturbance, the pressure of the indicated section varies a lot at different times, with the maximum pressure at $T/4$ and the minimum pressure at $3T/4$. Due to the combined effect of the swirl at the runner outlet and the low flow rate with a small opening, there is still a low-pressure zone in the center of the straight cone section of the draft tube, and the pressure value changes in line with the sinusoidal variation law of wave disturbance. However, the pressure distribution law of the indicated section is generally the same as that without wave disturbance, and the effect induced by wave disturbance is relatively small.

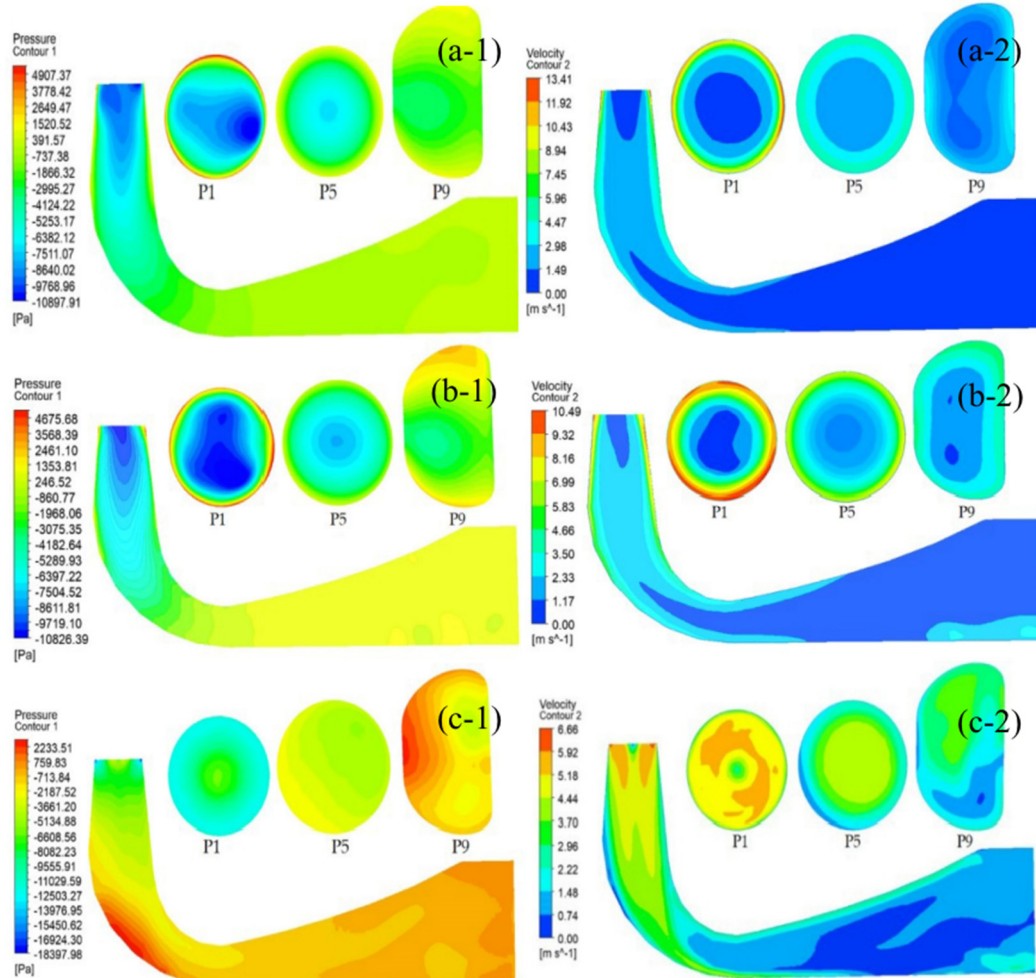

**Figure 4.** Static pressure and velocity distribution of draft tube with different opening without wave disturbance: (**a-1**) static pressure distribution of draft tube with $\alpha_0 = 6°$, (**a-2**) velocity distribution of draft tube with $\alpha_0 = 6°$, (**b-1**) static pressure distribution of draft tube with $\alpha_0 = 12°$, (**b-2**) velocity distribution of draft tube with $\alpha_0 = 12°$, (**c-1**) static pressure distribution of draft tube with $\alpha_0 = 24°$, (**c-2**) velocity distribution of draft tube with $\alpha_0 = 24°$.

As shown in Figure 6, under the running condition of $\alpha_0 = 12°$, with the increase in the opening of the guide vane, the inlet velocity of the draft tube gradually becomes even and stable, and the pressure gradient in the central area of the straight cone section slows down. Similar to the running condition of $\alpha_0 = 6°$, the maximum pressure of the draft tube appears in the area near the side walls of the draft tube inlet. However, a new vortex area appears at the end of the draft tube diffusion section. At the same time, the pressure value is smaller than that at $\alpha_0 = 6°$, the variation in pressure follows the sinusoidal variation law of wave disturbance, and the pressure distribution law affected by the wave disturbance is relatively small.

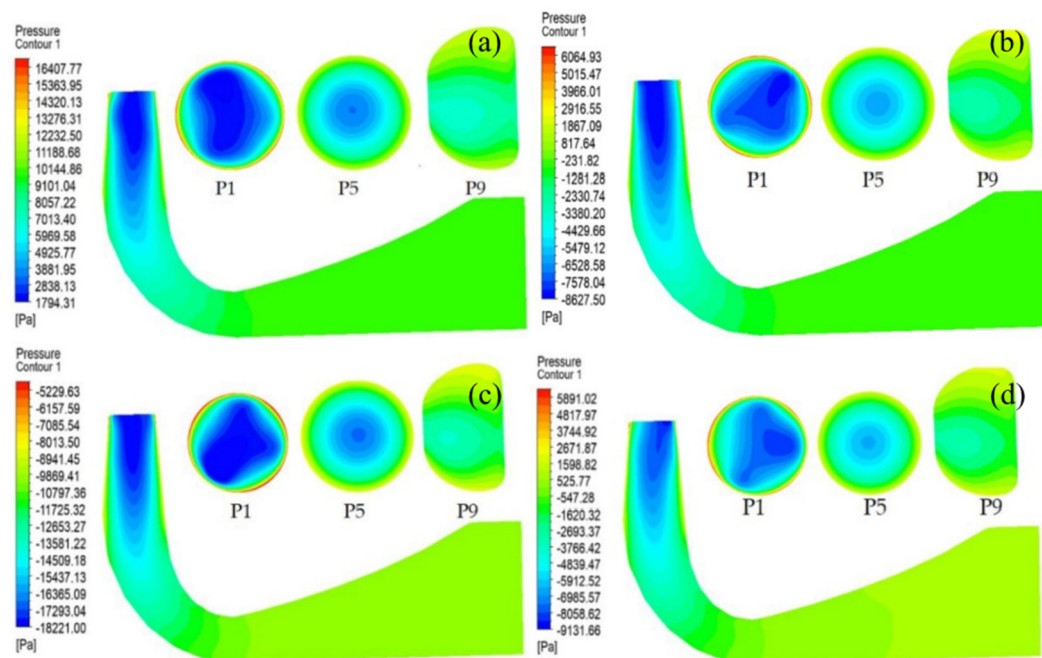

**Figure 5.** Static pressure distribution of the draft tube with $\alpha_0 = 6°$ under wave disturbance: (**a**) T/4, (**b**) T/2, (**c**) 3T/4, and (**d**) T.

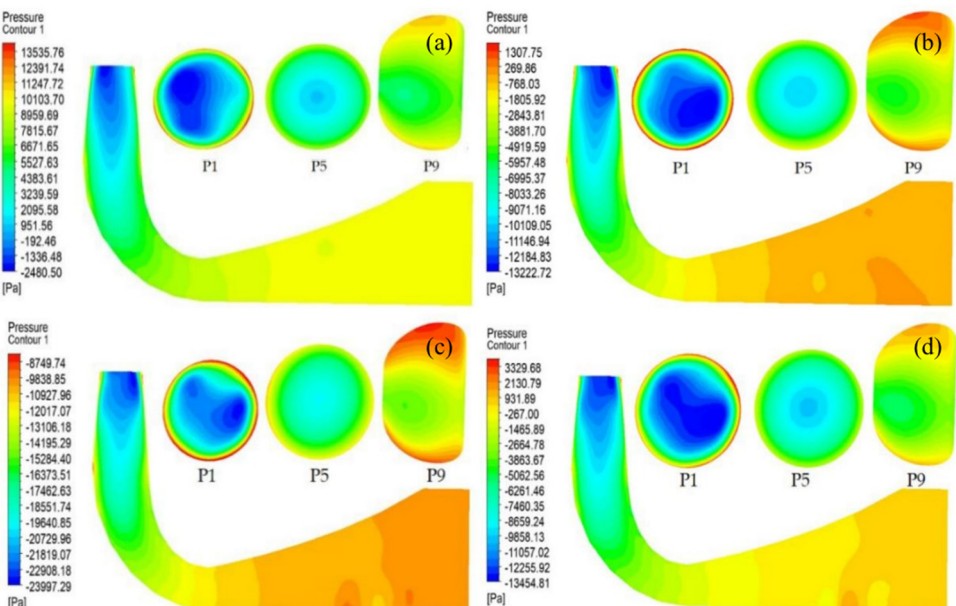

**Figure 6.** Static pressure distribution of the draft tube with $\alpha_0 = 12°$ under wave disturbance: (**a**) T/4, (**b**) T/2, (**c**) 3T/4, and (**d**) T.

As shown in Figure 7, for the situation with wave disturbance, under the running condition of $\alpha_0 = 24°$, with the increase in the opening of the guide vane, there is almost no pressure gradient in the center of the straight cone section of the draft tube. At the inlet of the draft tube, due to the large water flow rate and the elbow blocks of the water flow, a local high-pressure area appears at the elbow of the draft tube. In other areas, the pressure increases gradually from the inlet of the draft tube to the outlet of the draft tube. The variation in the pressure fits well with the sinusoidal variation law, and the pressure distribution law affected by the wave disturbance is relatively small. In the cloud chart of the pressure in the draft tube, an obvious pressure gradient can be observed in the center

of the straight cone section of the draft tube under the conditions of $\alpha_0 = 6°$ and $\alpha_0 = 12°$. In order to further explore the influence of wave disturbance on the draft tube pressure pulsation, it is necessary to analyze the pressure pulsation of each monitoring point in the frequency domain.

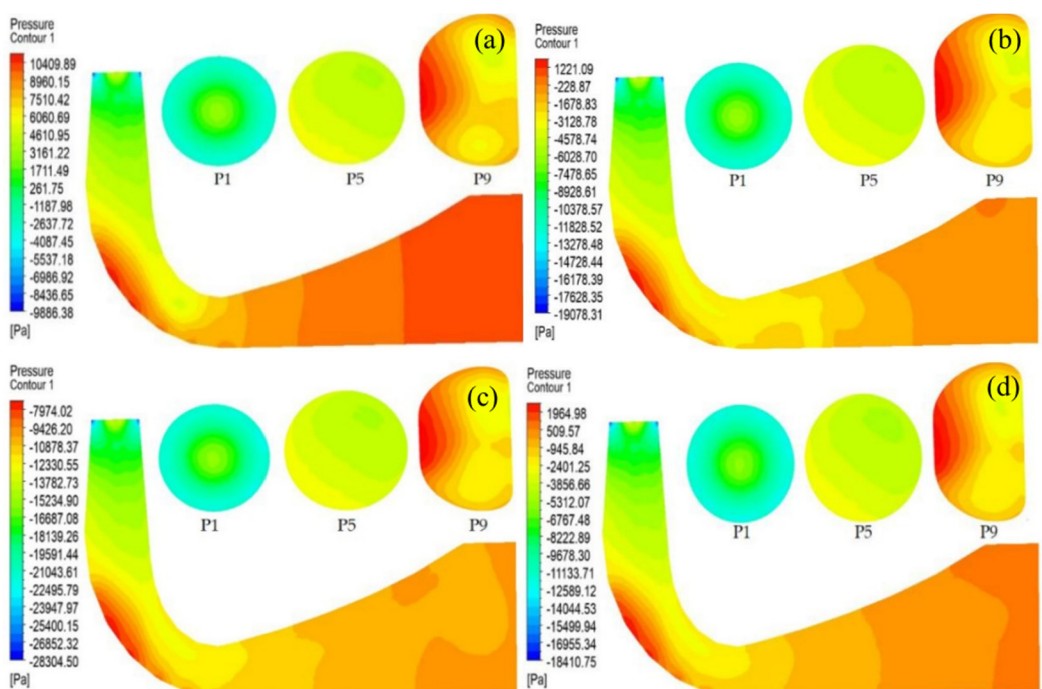

**Figure 7.** Static pressure distribution of the draft tube with $\alpha_0 = 24°$ under wave disturbance: (**a**) T/4, (**b**) T/2, (**c**) 3T/4, and (**d**) T.

### 4.2. The Peak Value of Pressure Pulsation

In order to describe the influence of the guide vane opening and wave disturbance on the draft tube pressure pulsation more intuitively and quantify the main characteristics of the draft tube pressure pulsation, the pressure pulsation coefficient peak $\Delta H'$ and pressure pulsation coefficient $C_p$ are introduced [30]:

$$\Delta H' = \frac{\Delta H}{H} = \frac{P_{\text{imax}} - P_{\text{imin}}}{\rho g H} \times 100\% \tag{10}$$

$$C_P = \frac{P_i - \overline{P}}{\rho g H} \times 100\% \tag{11}$$

where $\Delta H$ is the peak value of the pressure pulsation, $H$ is the water head, $P_{imax}$ and $P_{imin}$ are the corresponding maximum and minimum pressures at point $i$, $C_p$ is the dimensionless pressure pulsation coefficient, $P_i$ is the corresponding pressure at point $i$, and $\overline{P}$ is the average pressure over time.

In addition, the $Q$ criterion is adopted to identify the transient vortex structure [31], and the definition of this criterion is shown in Equations (12)–(14).

The governing equation of the velocity gradient tensor is as follows:

$$\lambda^3 - P\lambda^2 + Q\lambda - R = 0 \tag{12}$$

where $\lambda$ is the eigenvalue of the characteristic equation, and $P$, $Q$, and $R$ are the three invariants of the velocity gradient tensor. $Q$ is the second invariant, defined as

$$Q = \frac{1}{2}(\Omega^2 - S^2) \tag{13}$$

where $\Omega$ is the vorticity tensor, and $S$ is the strain rate tensor. $Q > 0$ indicates that vorticity is dominant in the flow field region, and $Q < 0$ indicates that the strain rate or viscous stress is dominant in the flow field region. In the Cartesian coordinate system, $Q$ can be written as

$$Q = -\frac{1}{2}\left[\left(\frac{\partial u}{\partial x}\right)^2 + \left(\frac{\partial v}{\partial y}\right)^2 + \left(\frac{\partial w}{\partial z}\right)^2\right] - \left(\frac{\partial u}{\partial y}\right)\left(\frac{\partial v}{\partial x}\right) - \left(\frac{\partial u}{\partial z}\right)\left(\frac{\partial w}{\partial x}\right) - \left(\frac{\partial v}{\partial z}\right)\left(\frac{\partial w}{\partial y}\right) \tag{14}$$

It can be seen from Figure 8a that when there is no wave disturbance, the amplitude of the pressure pulsation at each monitoring point decreases with the decrease in the opening. When the opening of the guide vane is $\alpha_0 = 24°$, the maximum pressure pulsation coefficient of the draft tube is obviously higher than that of $\alpha_0 = 6°$ and $\alpha_0 = 12°$. In addition, the difference between the maximum pressure pulsation coefficient at $\alpha_0 = 6°$ and that at $\alpha_0 = 12°$ is relatively small. The maximum of the pressure coefficient pulsation decreases slowly along the center line of the monitoring points in the straight cone section, which means that the pressure pulsation propagation has a small attenuation in the straight cone section of the draft tube. The maximum pressure pulsation coefficient is 1.22% at P1 under $\alpha_0 = 24°$, and the minimum pressure pulsation coefficient is 0.36% at P9 under $\alpha_0 = 6°$. It can be seen from the $Q$ criterion vortex distribution diagram that, due to the combined effect of the swirl at the runner outlet and the low flow rate with a small opening, the vortex is obvious in the center of the straight cone section of the draft tube both at $\alpha_0 = 6°$ and $\alpha_0 = 12°$. The influence range of the eccentric vortex is about 0.85 $D_2$ at $\alpha_0 = 6°$, and about 0.65 $D_2$ at $\alpha_0 = 12°$. The vortex dissipates at $\alpha_0 = 24°$.

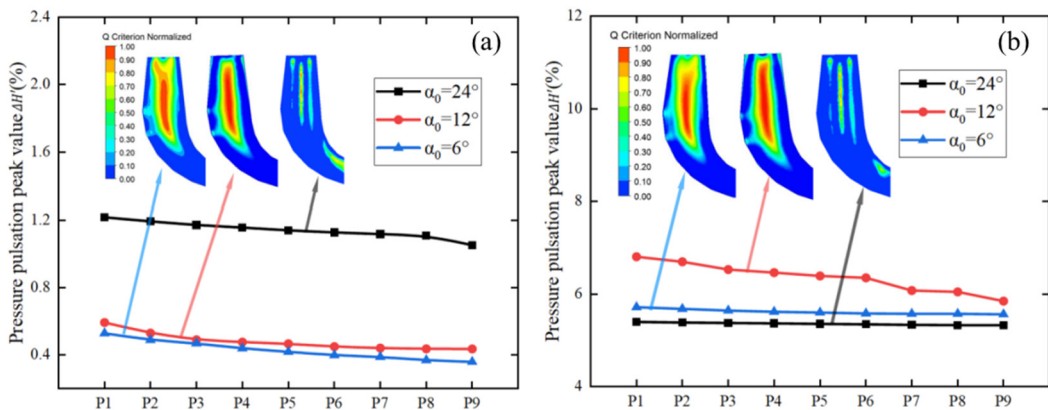

**Figure 8.** The maxima of pressure pulsation coefficient of monitoring points of the draft tube with different opening: (**a**) without wave disturbance, (**b**) wave disturbance.

It can be seen from Figure 8b that the wave disturbance has a negligible influence on the vortex distribution in the straight cone section of the draft tube; however, it has a significant influence on the amplitude of the pressure pulsation at each monitoring point. Under the running condition of $\alpha_0 = 12°$, the peak value of the maximum pressure pulsation coefficient at P1 is 6.82%. Under the running condition of $\alpha_0 = 24°$, the peak value of the minimum pressure pulsation coefficient at P9 is 5.33%. This means that the draft tube, as an important overcurrent component in the pump turbine, is too strong. In particular, the peak value of the maximum pulse coefficient with wave disturbance is more than five times larger than that without disturbance, which is a potential safety problem for the safe operation of the pump turbine.

### 4.3. Characteristics of Draft Tube Pressure Pulsation

Figure 9a,c,e show the frequency-domain results of the pressure pulsation at each measuring point without wave disturbance, where $f$ is the frequency, and $f_n$ is the frequency conversion of the runner. It can be seen that when the opening is small, the pressure amplitude is low, due to the low flow rate. The internal flow pattern of the pump turbine is

complicated in the case where the pump is running under a small opening; thus, there are more vortices and backflows, which may lead to more low-frequency components in the pressure pulsation under a small opening. When the opening of the guide vane reaches $\alpha_0 = 24°$, the vortex gradually dissipates, but the pressure amplitude still increases with the increase in the flow rate. There is a low-frequency pulsation of 1.95 Hz in the central basin of the straight cone section of the draft tube with different opening degrees, which is the main frequency of the draft tube pressure pulsation. Overall, the pressure coefficient $C_p$ increases obviously with the increase in the guide vane opening. In addition, under the same opening degree, the frequency components of all monitoring points are basically the same.

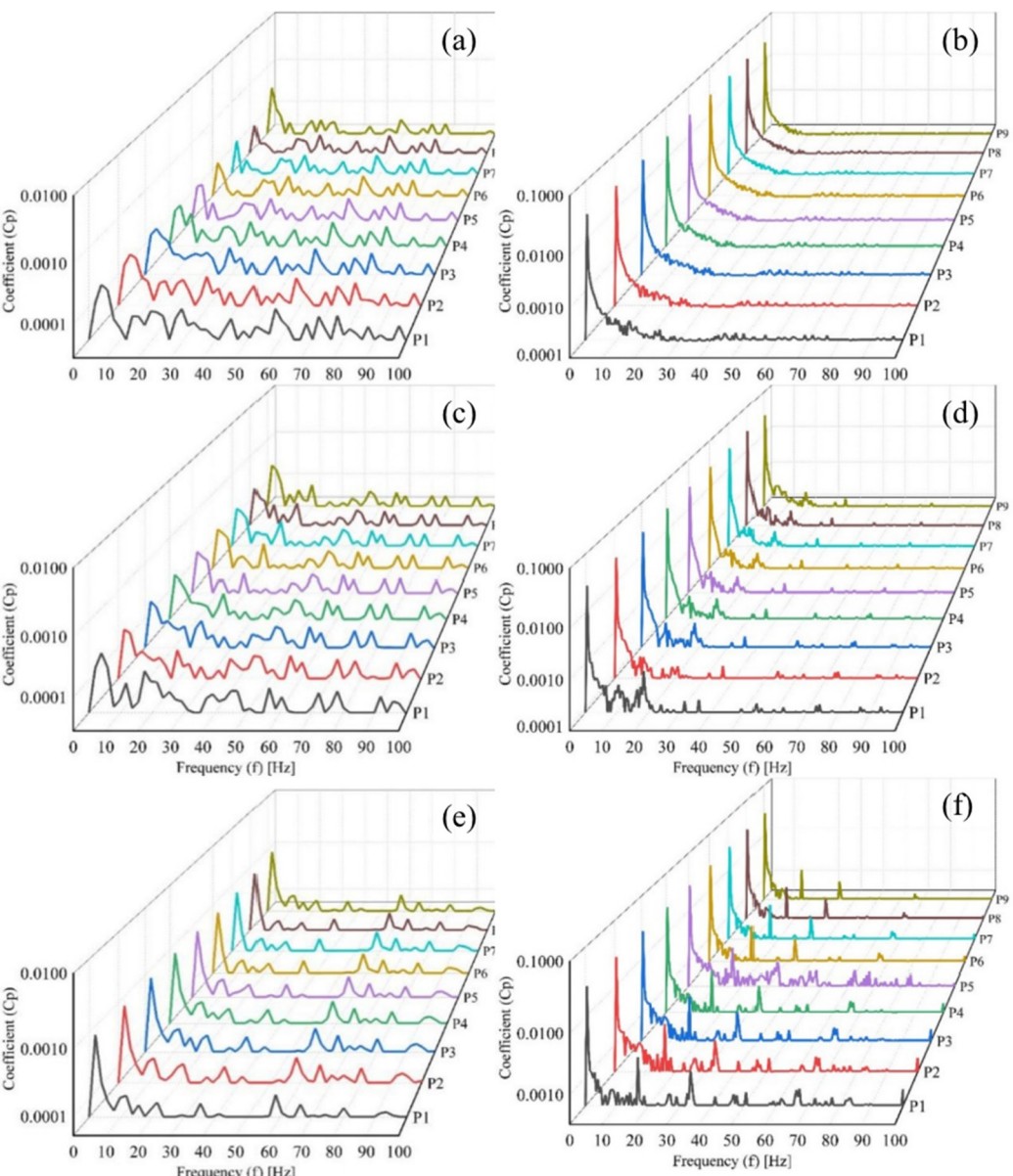

**Figure 9.** Frequency-domain diagram of the pressure pulsation: (**a**) $\alpha_0 = 6°$ without wave disturbance; (**b**) $\alpha_0 = 6°$ with wave disturbance; (**c**) $\alpha_0 = 12°$ without wave disturbance; (**d**) $\alpha_0 = 12°$ with wave disturbance; (**e**) $\alpha_0 = 24°$ without wave disturbance; (**f**) $\alpha_0 = 24°$ with wave disturbance.

Figure 9b,d,f show the frequency domain of the pressure pulsation at different measuring points for the situation with wave disturbance. Compared to the situation without wave disturbance, the pressure amplitude increases sharply, and the amplitude of the

original main frequency of 1.95 Hz at each monitoring point is no longer obvious. It is replaced by a low-frequency pulsation with a frequency of 0.5 Hz, which equals the frequency of the wave disturbance. The frequency spectrum of the pressure pulsation under each opening is different, and no significant regularity is observed. The frequency of the pressure pulsation under each opening changes, but the characteristics of the low frequency and high amplitude of the draft tube pressure pulsation remain.

### 4.4. Classification of Major Frequencies

As can be seen from Figure 10a,c,e, for the situation with no wave disturbance, with the increase in the guide vane opening, the overall pressure amplitude in the low-frequency region increases, and the pressure amplitude is $0.11\,f_n$ under all running conditions. As a similar unsteady pressure phenomenon, Yu et al. also confirmed that the main frequency of the pressure pulsation in the draft tube of a pump turbine unit was about $0.1\,f_n$ [32]. Under the running condition of $\alpha_0 = 6°$, the frequency corresponding to the maximum pressure amplitude of many upstream monitoring points becomes $0.23\,f_n$. Under the running condition of $\alpha_0 = 12°$, although the frequency corresponding to the maximum pressure amplitude of P1 is only $0.23\,f_n$, other monitoring points also have higher amplitudes at $0.23\,f_n$. When $\alpha_0 = 24°$, the pressure amplitude corresponding to $0.23\,f_n$ of each monitoring point is obviously smaller. According to [21], the rotating frequency of the vortex rope of a pump turbine is generally 1/3~1/5 of the rotating frequency, which is the main frequency of the tail race vortex rope. The rotating frequency of the runner is $1.0\,f_n$, which affects the whole draft tube. There is no change in the amplitude of the rotating frequency under each opening. With the increase in the guide vane opening, the amplitude of the pressure pulsation of the blade frequency of $7.0\,f_n$ decreases gradually. Both the rotating frequency and the blade frequency come from the dynamic and static interference between the runner and the guide vane. For other frequencies between $3.5\,f_n$ and $2.0\,f_n$, these frequencies generally come from the vortex and secondary flow, and their corresponding amplitudes are small.

As can be seen from Figure 10b,d,f, when there is wave disturbance and the guide vane opening is $\alpha_0 = 6°$ and $\alpha_0 = 12°$, there is still a high pressure amplitude at the main frequency of $0.23\,f_n$ of the vortex rope. With the increase in the opening of the guide vane, the high-amplitude pressure pulsation of the main frequency of $0.23\,f_n$ of the vortex rope gradually disappears. Because the monitoring point P1 is close to the runner, it is greatly affected by the dynamic and static interference, and has a high pressure amplitude at the rotating frequency of $1.0\,f_n$ of the runner. Overall, the maximum pressure amplitude of each monitoring point generally increases by 5~15 times, and the main frequency of the pressure pulsation is close to 0.5 Hz, which is the frequency of wave disturbance itself. In addition, there are several 0.5 Hz frequency-doubled, high-amplitude pressure pulsations, which indicates that the pressure waves generated by the waves propagate from the downstream side to the inside of the draft tube, and the superposition of the pressure pulsation signals produces more low-frequency, high-amplitude pressure pulsations.

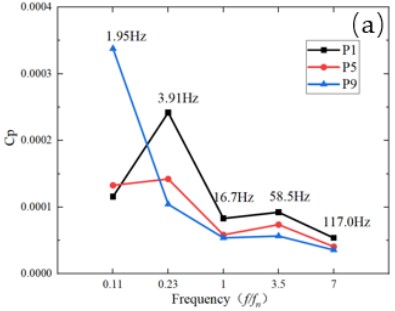 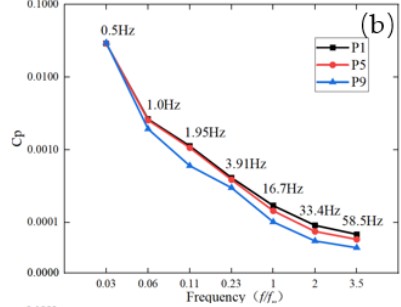

**Figure 10.** *Cont.*

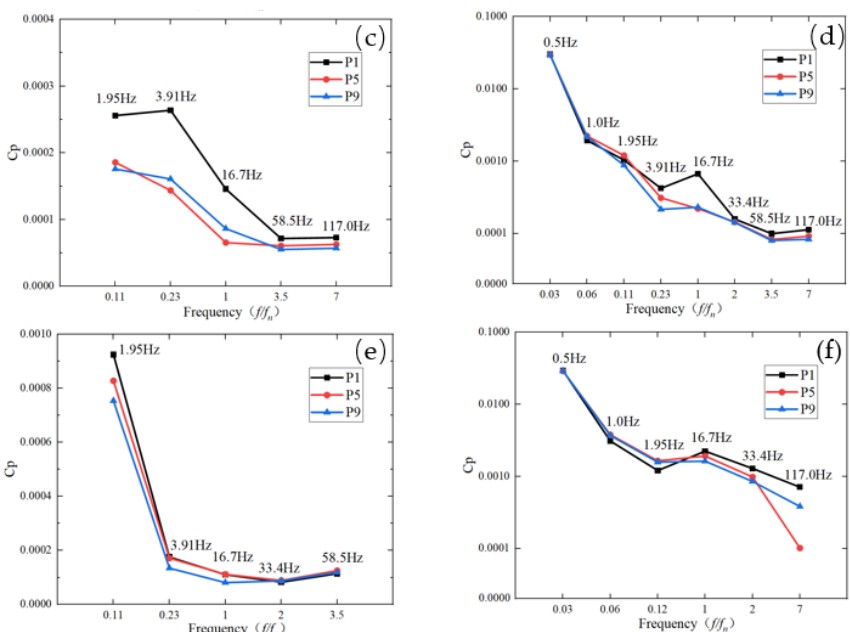

**Figure 10.** Main frequency types under different guide vane openings: (**a**) $\alpha_0 = 6°$ without wave disturbance; (**b**) $\alpha_0 = 6°$ with wave disturbance; (**c**) $\alpha_0 = 12°$ without wave disturbance; (**d**) $\alpha_0 = 12°$ with wave disturbance; (**e**) $\alpha_0 = 24°$ without wave disturbance; (**f**) $\alpha_0 = 24°$ with wave disturbance.

## 5. Conclusions

In this paper, we analyzed the transient hydraulic characteristics of the draft tube of a pump turbine under wave disturbance and discussed the influence of wave disturbance on the draft tube pressure pulsation under different guide vane openings. The concluding remarks are as follows.

Wave disturbance affected the pressure value in the draft tube. Its influence on the eccentric vortex rope in the straight cone section and the pressure distribution law in the draft tube was relatively small. For the situation without wave disturbance, the maximum pressure pulsation coefficient peak of the monitoring points appeared at $\alpha_0 = 24°$ with a value of 1.22%. For the situation with wave disturbance, the monitoring points' maximum pressure pulsation coefficient peak appeared at $\alpha_0 = 12°$ with a value of 6.82%. The maximum pressure pulsation at all monitoring points universally increased 5~15 times.

For the situation without wave disturbance, the central pressure amplitude of the straight cone section of the draft tube was small, and an obvious eccentric vortex appeared under the running condition of a small opening. With the increase in the opening, the eccentric vortex gradually dissipated. The flow rate was the main factor that affected the pressure pulsation. For the situation with wave disturbance, the central pressure amplitude of the straight cone section of the draft tube increased sharply. The wave disturbance and eccentric vortex were the main factors affecting the pressure pulsation.

For the situation without wave disturbance, the main frequency of the draft tube under each opening was $0.11 f_n$. The main frequency of the pressure pulsation in the eccentric vortex was $0.23 f_n$, showing the typical characteristics of a low frequency and high amplitude. For the situation with wave disturbance, the main frequency of the draft tube became 0.5 Hz, equaling the frequency of the wave disturbance. In addition, there were several 0.5 Hz frequency-doubled, high-amplitude pressure pulsations. The pressure pulsation frequency spectrum varied a lot, and the main frequency characteristics still existed.

The flow law and pressure pulsation propagation law of the pump turbine found in this study can provide a reference for the optimal design of seawater pumped storage units. In addition, it would be interesting to investigate the effect of wave disturbance on the flow

field and pressure pulsation of pump turbines under more complex operation conditions in further research.

**Author Contributions:** Conceptualization, J.H. and Z.M.; methodology, J.H., Q.W. and H.S. (Hongge Song); validation, B.C., Q.W. and H.S. (Hongge Song); formal analysis, J.H.; writing—original draft, B.C., J.H. and Q.W.; writing—review and editing, Z.M., Q.W. and H.S. (Hui Shen). All authors have read and agreed to the published version of the manuscript.

**Funding:** This research was funded by the Joint Funds of the Zhejiang Provincial Natural Science Foundation (LZJWZ22E090004).

**Institutional Review Board Statement:** Not applicable.

**Informed Consent Statement:** Not applicable.

**Data Availability Statement:** The numerical simulation data and experimental data used to support the findings of this study are available from the corresponding author upon request.

**Conflicts of Interest:** The authors declare no conflict of interest.

## Nomenclature

| | |
|---|---|
| $C_p$ | Coefficient of pressure fluctuation |
| $C_D$ | DES turbulence model constant |
| $D_1$ | Inlet diameter of runner (m) |
| $D_2$ | Outlet diameter of runner (m) |
| $f$ | Frequency (Hz) |
| $f_n$ | Rotation frequency (Hz) |
| $g$ | Acceleration of gravity (m/s$^2$) |
| $H$ | Head (m) |
| $K$ | Production term of turbulent kinetic energy |
| $K_z$ | Pressure sensitivity coefficient |
| $l_p$ | Prototype length (m) |
| $l_m$ | Model length (m) |
| $n$ | Rotational speed (r/min) |
| $n_{11}$ | Specific speed (r/min) |
| $p$ | Static pressure (Pa) |
| $Q_{11}$ | Specific flow rate (m$^3$/s) |
| $S$ | Generalized source term of momentum equation |
| $S_i$ | Invariant measure of the strain rate |
| $Z$ | Elevation of the free surface (m) |
| $\alpha_0$ | Guide vane opening (degree) |
| $\sigma_k$ | Constant of the turbulence model |
| $\rho$ | Density (kg/m$^3$) |
| $\mu$ | Dynamic viscosity (Pa s) |
| $\varphi$ | Velocity potential |
| $\Gamma$ | Fluid parameters |
| $\Omega$ | Vorticity tensor |
| $\eta$ | Efficiency (%) |
| $\lambda$ | Eigenvalue of the characteristic equation |
| $\lambda_k$ | Scale length of wave model |
| $\lambda_p$ | Scale of wave pressure |
| $\lambda_f$ | Scale of wave frequency |
| $\Delta H$ | Peak value of pressure pulsation (m) |
| $\Delta H'$ | Relative pressure pulsation amplitude (%) |
| CFD | Computational fluid dynamics |
| DES | Detached eddy simulation |
| FFT | Fast Fourier transform |
| LES | Large eddy simulation |
| RANS | Reynolds-averaged Navier–Stokes |
| SST | Shear stress transport |
| SIMPLEC | Semi-Implicit Method for Pressure-Linked Equations-Consistent |

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
