# Peer review of "Numerical Study of the Internal Fluid Dynamics of Draft Tube in Seawater Pumped Storage Hydropower Plant"

_sustainability, doi:10.3390/su15108327_

Round 1

Reviewer 1 Report

The results presentation is interesting and the discussion on results is appropriate and needs some corrections. It is recommended to publish it after minor revision

Minor corrections

(1)   Line 78, ‘vortex belt’ should be ‘vortex rope’.

(2)   Line 138, ‘is lager’ should be ‘larger’.

(3)   Some references should be enclosed in parentheses, such as line 170, ‘28’ should be ‘[28]’.

(4)   Line 219, ’of’ should be canceled.

(5)   Line 291, ‘of ‘should be canceled.

(6)   Line 292, ‘as’ should be ‘has’.

(7)   Line 328, ‘shows’ should be ‘show’.

(8)   Line 332, ‘is’ should be canceled.

(9)   Line 395, ‘affecting’ should be ‘affects’.

(10) Line 405, ‘is varies’ should be ‘varies’.

Some suggestions:

(11)  Please suggest the functions of the workstations or desktops the authors use for simulation tasks.

(12)  What is the software used for CFD?

(13)  Why did this article choose the DES model for calculation and what are the advantages?

(14)  Nomenclature part should be included.

(15)  Result and discussion can be improved from the application point of view.

good 

Reviewer 2 Report

The paper does a fluid dynamics numerical simulation of draft tube in seawater pumped storage hydropower plant and analyzes the influence of wave disturbance on draft tube pressure pulsation under different guide vane openings. Despite not introducing any new method, I believe the paper results may be a good reference for future works.

Specific comments:

- Fit the size of brackets and parenthesis to their contents in equations (e.g. (2)-(4)).

- Do not indent text after equations in the same phrase (e.g. “where” after (4)).

- Line 169: what does 6s27 stand for? Maybe 6.27s is clearer, if it is the correct equivalent.

- Line 187: do not use “*” for multiplication. Use “x” (\times in latex) instead or nothing. An “*” induces convolution.

- Variables must be in italic (e.g. H, t, T in line 187, Q, D_2, H, in line 199).

- Equations must be part of the text (e.g. (8) and (9) would be better written as “according to … (8) … (9) where…”).

- Section 4.4 title is orphan. Take care in the final version.

- Figure 9 is missing.

The paper needs some language review (e.g. “frequence” should be “frequency” in Figure 10, “… frequency spectrum is varies a lot…” should be “… frequency spectrum varies a lot…”).

Reviewer 3 Report

This work deals with a numerical study that focuses on analyzing aspects influenced by wave disturbances in suction tubes in pump turbines. The main result of this study is how expressive is the influence of perturbation by waves when compared without them, that is, something around up to 15 times. I understand that these results may be useful for other researchers in the field of turbines (wind turbines, for cars, steam, hydraulics, among others), however, for acceptance of this work, the following corrections are required:

1 - In line 123 correct ... "9monitoring" points;

2 - Measurement units must be separated from numbers, note for example in Table 1 that it has "552.2mm";

3 - in lines 154 and 155 the physical properties are damaged in the text I received, and more, it is necessary to describe the units of measurement;

4 - it would be important to describe why " ... the maximum number of iterations in each time step is set to 30 ... ", in other words why not less or more? What is the reason for this choice? What about numerical accuracy or machine time (computation)? (see line 162);

5 - in line 170 has "can be expressed as28:", what is this 28?

6 - In line 171, because equation (5) was used, is there any reference that supports such use?;

7 - In line 179 has "wave scale are as follows 29:", what is this 29? is it a reference? Is that how you refer to it in the text? see that it's the same thing with "28" on line 170;

8 - na linha 129 tem a Tabela 1 e na linha 184 novamente Tabela 1???

9 - again in Table 1 of line 184, there is a lot of stuff "stuck" that hinders the clear evaluation of the article, note that in this table there are "alpha0", "n11(r/min)". It also has "Q11(m3/s)" is it not possible to put that "3" superscript??

10 - in Figure 2, line 204, everything is stuck together for example, alpha_0=24º, you have to put a space between alpha, the equal sign and the numerical value; this type of error is very common in this text and it is not technical and good reading for other researchers;

Round 2

Reviewer 3 Report

The authors complied with all claims, and therefore, I understand that the article can be accepted for publication.